# Compressional pathways of α-cristobalite, structure of cristobalite X-I, and towards the understanding of seifertite formation

Ana Černok[1,†], Katharina Marquardt[1], Razvan Caracas[2], Elena Bykova[1], Gerlinde Habler[3], Hanns-Peter Liermann[4], Michael Hanfland[5], Mohamed Mezouar[5], Ema Bobocioiu[2] & Leonid Dubrovinsky[1]

In various shocked meteorites, low-pressure silica polymorph α-cristobalite is commonly found in close spatial relation with the densest known $SiO_2$ polymorph seifertite, which is stable above ~80 GPa. We demonstrate that under hydrostatic pressure α-cristobalite remains untransformed up to at least 15 GPa. In quasi-hydrostatic experiments, above 11 GPa cristobalite X-I forms—a monoclinic polymorph built out of silicon octahedra; the phase is not quenchable and back-transforms to α-cristobalite on decompression. There are no other known silica polymorphs, which transform to an octahedra-based structure at such low pressures upon compression at room temperature. Further compression in non-hydrostatic conditions of cristobalite X-I eventually leads to the formation of quenchable seifertite-like phase. Our results demonstrate that the presence of α-cristobalite in shocked meteorites or rocks does not exclude that materials experienced high pressure, nor is the presence of seifertite necessarily indicative of extremely high peak shock pressures.

[1] Bayerisches Geoinstitut, Universitätsstrasse 30, 95447 Bayreuth, Germany. [2] CNRS, Laboratoire de Geologie de Lyon, UMR 5276, Université Claude Bernard Lyon 1, Ecole Normale Supérieure de Lyon, 46, allée d'Italie, 69364 Lyon, France. [3] Department of Lithospheric Research, University of Vienna, Althanstrasse 14, 1090 Vienna, Austria. [4] Photon Sciences, Deutsches Elektronen-Synchrotron (DESY), Notkestraße 85, 22607 Hamburg, Germany. [5] European Synchrotron Radiation Facility (ESRF), 6 Rue Jules Horowitz, 38000 Grenoble, France. † Present address: The Open University, School of Physical Sciences, Walton Hall, Milton Keynes MK7 6AA, UK. Correspondence and requests for materials should be addressed to A.Č. (email: ana.cernok@open.ac.uk) or to R.C. (email: razvan.caracas@ens-lyon.fr).

The high-temperature, low-pressure polymorph of silica—α-cristobalite—has a subordinate and rather exotic terrestrial occurrence among silica phases (for example, quartz), but it is observed as the predominant $SiO_2$ polymorph in various planetary materials—for example, planetary dust[1], various meteorites[2] or lunar rocks[3]. The response of the α-cristobalite found in meteorites to the high pressure (and high-temperature) conditions encountered during an impact still puzzles scientists because of its common spatial association with high-pressure mineral assemblages. On the one hand, α-cristobalite is found in meteorites that lack any high-pressure silica polymorphs albeit the rest of the minerals indicate peak shock pressures in excess of 10 GPa and high temperatures, at which coesite or stishovite are expected[4,5]. On the other hand, it is documented alongside all the natural occurrences of seifertite, a post-stishovite high-pressure polymorph of $SiO_2$ only found in heavily shocked (25 GPa or higher) meteorites[5–8]. According to these observations, α-cristobalite seems to be stable at variable pressure conditions (from ambient to more than 25 GPa), thereby not recording the peak transient pressure as the other associated phases. Triggered by this curious behaviour and later by interesting petrographic observations, particularly by the coexistence of seifertite and α-cristobalite, the behaviour of α-cristobalite at high pressures was examined by numerous experimental[9–18] and theoretical studies[19–21] for more than two decades. While they all indicated abundant polymorphism at elevated pressures, no consensus has emerged on what is the exact sequence of the pressure-induced transformations in α-cristobalite. The results are particularly discrepant when comparing different forms of starting materials (for example, powder or single crystal) and different levels of hydrostaticity[9,10,12,13,16]. Experiments[9,12,13,16,17] carried out at non-hydrostatic conditions (that is, in solid pressure-transmitting medium or without any medium) using powders as starting material report the transformation of α-cristobalite, via cristobalite-II and cristobalite X-I, to seifertite—the α-$PbO_2$-type silica—near 30 GPa. As such, they provide evidence that seifertite can form at pressures much lower than expected from its thermodynamic equilibrium (in excess of 80 GPa[22,23]) thus bypassing the formation of thermodynamically expected stishovite and $CaCl_2$ polymorph. A similar path (α→II→X-I) was observed on a single crystal studied in hydrostatic (liquid pressure medium) or quasi-hydrostatic (solidified but very soft, Ne particularly, pressure-transmitting media) experiments at lower pressures, up to ∼12 GPa (ref. 10). However, a single-crystal α-cristobalite has not been studied at pressures exceeding the formation of cristobalite X-I (∼11 GPa). Therefore, the structure of this 'bridging phase' X-I that forms in either quasi- (for example refs 10,18) or non-hydrostatic, (for example refs 12,13) conditions remained elusive until now. In terms of studying the structure and behaviour of materials at elevated pressures employing cutting-edge experimental procedures, silica remains one of the most challenging compounds due to its relatively low X-ray scattering factor, as well as its abundant polymorphism.

The only way to address the curious coexistence of cristobalite and seifertite, as well as the large discrepancies regarding pressures estimates in shocked meteorites based on the presence of seifertite, is to systematically study the behaviour of α-cristobalite under pressure with the focus on the formation of cristobalite X-I and α-$PbO_2$-type silica that is, seifertite. Here, we examined the response of various forms of α-cristobalite upon cold compression at different levels of hydrostatic stresses. The results reveal that the mechanism which α-cristobalite accommodates under compression is even more sensitive to stress conditions than previously anticipated[9,10,12–18]. By clarifying the stability field of cristobalite X-I and revealing its structure, we gained insight into the long disputed transition mechanism α→II→X-I→α-$PbO_2$. We provide clear evidence that the formation of seifertite at pressures much lower than its thermodynamic equilibrium is restricted to a non-hydrostatic environment (at room temperature). This explains the seemingly controversial observations regarding its natural coexistence with α-cristobalite.

## Results

**An overview of acquired data sets**. High-pressure experiments were carried out on both single crystals and powders (Supplementary Fig. 1). All single-crystal experiments were conducted in quasi-hydrostatic conditions, using a pressure-transmitting medium, whereas powders were also studied in non-hydrostatic environment. Starting material in form of α-cristobalite was compressed using diamond-anvil cells (DAC). *In situ* high-pressure data were obtained by Raman spectroscopy and synchrotron based single-crystal X-ray diffraction (SCXRD). The experimentally determined structure was then used in *ab initio* calculations to examine ordered structural models and to compute the Raman spectra and unit-cell parameters at various pressures. After performing high-pressure experiments, recovered samples were additionally investigated by powder X-ray diffraction (PXRD) and transmission electron microscopy (TEM).

**Effect of hydrostaticity on cristobalite transformation path**. We find that the mechanism which α-cristobalite adopts as response to compression strongly depends on the stress conditions. The effect is illustrated in Fig. 1: three α-cristobalite single crystals of different size were loaded inside the same pressure chamber of a DAC in neon pressure-transmitting medium. Upon compression, the smallest and the thinnest crystal (∼10 μm thickness, 'C3') that does not bridge between the anvils, remains fully surrounded by very soft neon experiencing (almost) hydrostatic environment, and thus retains the structure of α-cristobalite up to at least ∼15 GPa (Supplementary Table 1 and Supplementary Fig. 2). On the other hand, the two thicker crystals ('C1' & 'C2') that touch the diamond anvils upon compression, undergo a displacive phase transition to cristobalite-II near 1 GPa. The same crystals transform to cristobalite X-I just above ∼10 GPa, in agreement with earlier findings[10]. The Raman spectra of cristobalite X-I can be followed at least up to ∼80 GPa under compression (the maximum pressures studied here, Supplementary Fig. 3). After recovering cristobalite X-I from high-pressure experiments, it transformed back to the initial α-cristobalite, the structure of which was confirmed by Raman spectroscopy, TEM and X-ray diffraction (Supplementary Fig. 7). In contrast to single crystals, all compression experiments carried out on powders, regardless on the presence of the pressure-transmitting medium, showed identical transformation path, ultimately resulting in the formation of seifertite-like material (see below).

**α→II→X-I transitions**. The Raman spectra collected on α-cristobalite shown in Fig. 1 were complemented by X-ray diffraction analyses (Supplementary Fig. 2), which lead to three important conclusions: (1) α-cristobalite can transform to cristobalite-II at pressures higher than ∼1.5 GPa (in BA-DAC1 α-cristobalite was last observed at ∼3 GPa, before it transformed to cristobalite-II); (2) it is impossible to distinguish α-cristobalite from cristobalite-II above ∼2 GPa solely based on the main Raman $A_g$ mode position, because the doublet characteristic of cristobalite-II merges into one intense peak upon further

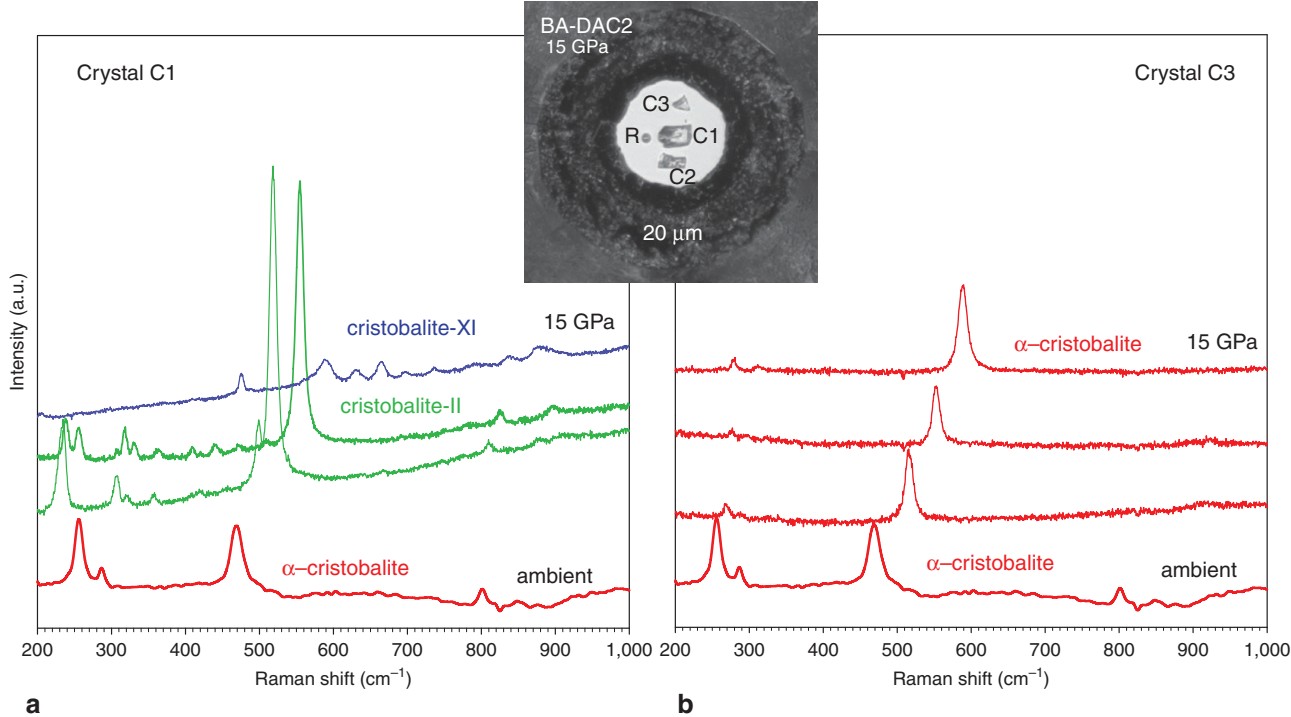

**Figure 1 | *In situ* Raman spectra of cristobalite at high pressures.** Spectra collected on two different single crystals of α-cristobalite at 1.1, 4.6, 8.6 and 15.0 GPa and at room temperature. Inset shows the DAC pressure chamber seen at 15.0 GPa: R stands for ruby; C1, C2 and C3 for different cristobalite crystals. In quasi-hydrostatic conditions (**a**) the crystal C1 starting as α-cristobalite (red) transforms via intermediate cristobalite-II (green) to cristobalite X-I (blue). However, in the case of high hydrostaticity (**b**), the smallest crystal C3 retains its initial structure of α-cristobalite (red). All spectra are unsmoothed, the background has been subtracted. Ar⁺ laser was used (514 nm excitation wavelength) with ∼0.8 W incident beam power.

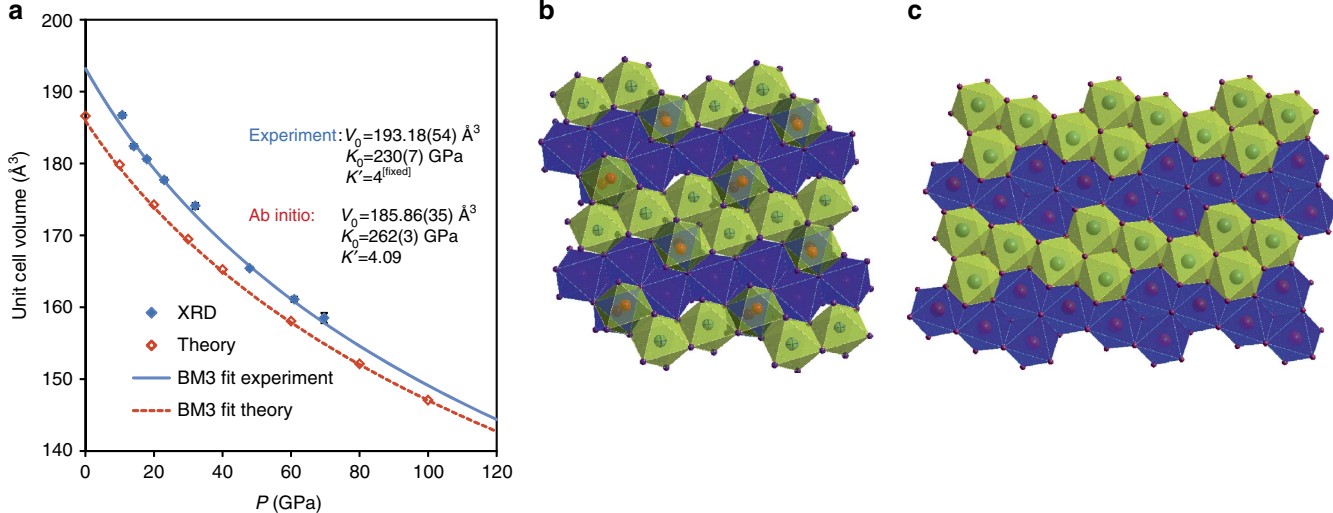

**Figure 2 | Structure and compressibility of cristobalite X-I.** (**a**) Third-order Birch-Murnaghan Equation of State fit to the *P*–*V* data obtained in experiment (blue line) and in *ab initio* calculations (red line). Exact unit-cell parameters obtained experimentally and theoretically are given in Supplementary Tables 3 and 4. (**b**) [010] projection of the structure of cristobalite X-I as obtained experimentally. The experimental structure contains half-occupied octahedral positions (Si3) shown by red spheres, and fully occupied octahedra (Si1 and Si2) represented by all other spheres, see Table 1. The blue and the green octahedra represent different levels in the structure. A row of three fully occupied octahedra surrounded by two half-occupied octahedra at each side form the '5 × 2 × 2 skeleton'. (**c**) The ordered structure used for model in *ab initio* calculations. The theoretical structure contains 4 × 4 arrangement of the fully occupied octahedra (Supplementary Table 6). The blue and the green octahedra represent different levels in the structure.

compression, showing the same pressure band shift as that of the α-cristobalite $A_g$; (3) compression of α-cristobalite up to ∼15 GPa, far outside its stability field ('overpressurized') in BA-DAC1, and in crystal C3 of BA-DAC2 is consistent with the observation

by Dera *et al.*[10]. The observed α→II transition above 3 GPa (BA-DAC1), as well as the coexistence of α-cristobalite with cristobalite-II and with cristobalite X-I (BA-DAC2) to at least ∼14 GPa strongly suggest that the main cause for the onset of the

**Table 1 | Structure of cristobalite X-I at 14.1 GPa.**

| Atom | Site | Occupancy | x | y | z | U [Å$^2$] |
|------|------|-----------|-----------|--------------|-----------|-------------|
| Si1  | 2d   | 1         | 0.5       | 0            | 0         | 0.0420(11)  |
| Si2  | 4e   | 1         | 0.8601(2) | 0.49817(16)  | 0.1207(3) | 0.0314(11)  |
| Si3  | 4e   | 0.5       | 0.7719(6) | 0.9939(4)    | 0.7659(8) | 0.0405(12)  |
| O1   | 4e   | 1         | 0.7904(4) | 0.2357(6)    | 0.5459(6) | 0.0314(11)  |
| O2   | 4e   | 1         | 0.9555(5) | 0.7244(9)    | 0.7098(6) | 0.0389(12)  |
| O3   | 4e   | 1         | 0.8313(5) | 0.7314(8)    | 0.3311(6) | 0.0369(11)  |
| O4   | 4e   | 1         | 0.9226(5) | 0.2733(8)    | 0.9162(6) | 0.0391(11)  |

The CCDC number assigned to this structure is 1491299.
Atomic parameters of the structure determined by single-crystal X-ray diffraction. Total number of unique reflexions 207, $R_{int}$ = 4.6%, $R_1$ = 9.3%, wR2 = 0.2564, GooF = 1.127, 28 parameters refined, 0 restrictions. Lattice parameters: $a$ = 6.5700(23) Å, $b$ = 4.0633(6) Å, $c$ = 6.8345(45) Å, $\beta$ = 98.001(56)°, $V$ = 180.69(15) Å$^3$. Composition SiO$_2$, Z = 8, $\rho$ = 4.416 g cm$^{-3}$. Space group $P2_1/n$ (Number 14, $b$-unique, setting 1). U is displacement paramter. Tranformation matrix to $P2_1/c$: [100 0–10 −10–1].

$\alpha \rightarrow$ II transition is the change in stress conditions. To test the hypothesis by Dera et al.[10] that the 'overpressurization' is accommodated by the very rapid compression far beyond the $\alpha \rightarrow$ II transition pressure ($\sim$1.5 GPa) we conducted an experiment (EXP 5, Supplementary Figs 1, 2 and 7) applying instant pressure-increase to $\sim$9 and then $\sim$12 GPa. We still observed formation of cristobalite-II and X-I. Moreover, crystal C3 (experiment BA-DAC2) evidenced 'overpressurization' of $\alpha$-cristobalite at the same rate of compression simultaneously with $\alpha \rightarrow$ II $\rightarrow$ X-I transition occurring in C1 and C2. The X-ray diffraction analysis of the 'overpressurized' $\alpha$-cristobalite was measured at 1.55 and 14.1 GPa. It shows nice diffraction pattern of a single crystal with no indication of twinning, resulting in accurate structural refinement (Supplementary Table 1).

**Cristobalite X-I.** Cristobalite-II is observed up to $\sim$11 GPa, after which it undergoes a first-order transition to cristobalite X-I, consistent with previous observations[10,15]. The abrupt change in Raman spectra suggests that the structure of the X-I phase is quite distinct from its $\alpha$- and cristobalite-II precursors (Fig. 1 and Supplementary Fig. 3). Upon compression the Raman spectra of cristobalite X-I can be followed up to $\sim$80 GPa and on decompression the Raman modes of cristobalite X-I can be followed down to $\sim$2 GPa, below which it transforms back to the starting $\alpha$-cristobalite (Supplementary Figs 3 and 7).

Even the most recent study regarding the formation and stability of cristobalite X-I[18] was based on powder-diffraction data, which were insufficient for structure determination. Our new single-crystal data reveal the structure of cristobalite X-I shown in Fig. 2 (see also Table 1).

The polymorph cristobalite X-I belongs to the family of high-pressure silica phases comprising of a distorted hexagonal close-packed array of oxygen ions in which silicon atoms fully or partially occupy octahedral sites[24–26], being sixfold coordinated to oxygen (Fig. 2b and Supplementary Fig. 4a). The concept of randomly distributed Si cations over half of the octahedral sites was proposed in 1978 by Liu et al.[24] as the modified niccolite structure. The existence of such a phase has been proposed in a powder diffraction study by Prakapenka et al.[13], but it has never been observed by a single-crystal study that can reveal the critical details of its structure. A unit-cell of cristobalite X-I contains three distinct octahedral sites (Table 1): Si1 and Si2 are fully occupied, with respectively 2 and 4 multiplicities; Si3 is half-occupied with multiplicity of 4. The Si1 and Si2 are less distorted, with Si–O bond lengths and SiO$_6$ octahedral volume similar to stishovite at similar pressures (Supplementary Table 2). The SiO$_6$ octahedron around Si3 is considerably larger and significantly more distorted than around Si1 and Si2 (Supplementary Table 2); it is similar to that of seifertite at ambient conditions.

Earlier theoretical simulations[19–21,25] did not predict the structure of cristobalite X-I observed in our study. Molecular dynamics simulations on cristobalite supercells[21] obtained a compact structure with SiO$_4$ tetrahedra at high pressures, which eventually transformed into stishovite under further compression. Metadynamics simulations starting with quartz yield metastable kinked octahedral chains under compression[20]. Various other chain-like configurations have been proposed previously for the high-pressure forms of silica[25,26]. However, our diffraction data suggest that the chains of octahedra in cristobalite X-I are linked by two pairs of neighbouring octahedra (Fig. 2b and Supplementary Fig. 4). On average at the scale of a crystal, the arrangement can be described as 5 × 2 × 2 ordering of octahedra, containing five octahedra in a row being connected with two pairs, each with one half-occupied connecting octahedra (in this manner, seifertite would be described as 2 × 2 × 1 ordering). However, at the local scale only one of the two octahedral sites is occupied. The resulting actual structure may be considered as random combination of 4 × 4 (Fig. 2c and Supplementary Fig. 4c) and 5 × 3 (Supplementary Fig. 4b) octahedra in adjacent rows. We performed the *ab initio* calculations using density-functional theory (DFT) in local-density approximation (LDA) to study these two ordered end-member configurations, as well as their random combination. The calculations show that the 4 × 4 end-member term is lower in enthalpy by 26 meV per formula unit at 10 GPa than the 5 × 3 one. The difference in enthalpy decreases almost linearly to 8 meV per formula unit at 100 GPa, but it is small enough to be overcome by the thermal energy at ambient conditions and could therefore explain experimentally observed disorder.

Next, we compute the Raman spectra for several ordered supercells (Supplementary Figs 5 and 6) at several pressures (10, 40 and 100 GPa) and compare them with our experimental results. First, we consider the two end-member terms, 4 × 4 and 5 × 3. The calculated Raman spectra for the supercell 4 × 4 arrangement show remarkable agreement with the experiment (within 5 cm$^{-1}$ or better) throughout the entire studied pressure range. The 5 × 3 arrangement has a series of mismatched peaks (Fig. 3). This suggests that in the disordered phase the 4 × 4 arrangement is dominant; however based on diffraction data and energetic considerations, the 5 × 3 cannot be entirely ruled out. Indeed, theoretical Raman spectra for two supercells containing equal amounts of the 4 × 4 and 5 × 3, ordered along the x or the y axis show a more realistic image of the spectra, with general good agreement to measurements (Supplementary Fig. 6).

The theoretical study did not reveal any sign of dynamical instabilities of cristobalite X-I at pressures up to at least 100 GPa. A third-order Birch–Murnaghan equation of state[27] (BM3 EOS)

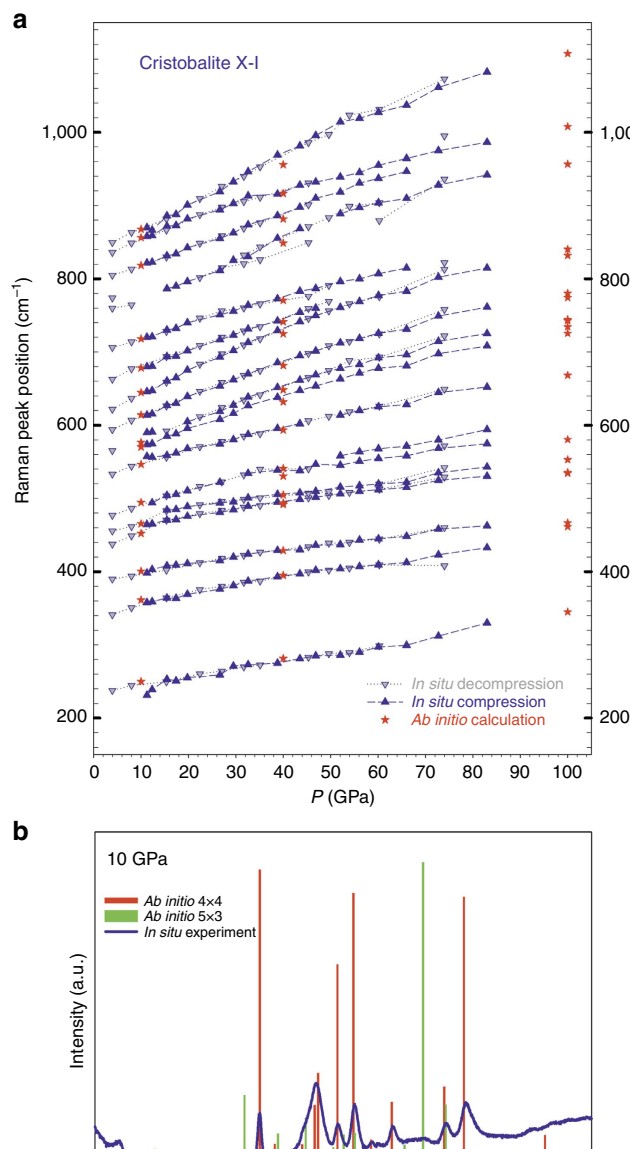

**Figure 3 | Pressure Raman spectra of cristobalite X-I.** (**a**) Pressure dependence of the Raman modes in cristobalite X-I at ambient temperature obtained experimentally (blue triangles 'up' for compression and 'down' for decompression) and simulated by *ab initio* calculations for the $4 \times 4$ arrangement (red stars). The lines are guides for enhanced readability. (**b**) Comparison of the Raman experimental spectrum (blue continuous line) and theoretical spectra for the $4 \times 4$ (red vertical bars) and $5 \times 3$ (green vertical bars) arrangements at 10 GPa.

fit to the experimental $P$–$V$ data (Fig. 2) yields $V_0 = 193.18(51)$ Å³ and $K_0 = 230(7)$ GPa with fixed $K' = 4$. When constructing the plot of the normalized stress[27], $F_E$, versus Eulerian strain, $f_E$, from our $P$–$V$ data starting from this value of $V_0 = 193.18(51)$ Å³, we find that it reflects linearity of data consistent with a BM3 fit, yielding values of the bulk modulus, $K_0$, of about 228 GPa and its pressure derivative $K'$ of 4.3 comparable to $K_0 = 230(7)$ GPa obtained by fitting. Theoretical results ($K_0 = 262(3)$ and $K' = 4.09$) are in reasonable agreement with experimental observations, especially taking into account that in calculations supercells with fixed distribution of silica

among octahedra were employed (while experimental data imply random distribution of Si in one of the octahedra). The bulk modulus of cristobalite X-I is lower than that of any other high-pressure $SiO_2$ phase containing silicon in octahedral coordination[23]—stishovite (300–310 GPa), $CaCl_2$ (320–330 GPa) or seifertite (320–330 GPa). This agrees with the fact that the structure of cristobalite X-I contains significant amount of partially occupied octahedra. High-pressure behaviour of cristobalite X-I is compared to selected tetrahedrally- and octahedrally-coordinated silica phases up to megabar region (Fig. 4), using available literature data and by DFT LDA calculations. The volume (per formula unit) of cristobalite X-I should become smaller than that of other silica phases with sixfold coordinated silicon at megabar pressures (Fig. 4). Thus, it cannot be entirely ruled out that cristobalite X-I or some other similarly disordered silica phase becomes stable at high pressures and certain temperatures. In addition, we computed energy–volume curves of α-cristobalite, quartz, cristobalite X-I and seifertite, using more accurate generalized-gradient approximation functional (GGA) at same pressures (inset in Fig. 4 and Supplementary Fig. 8). The $E$–$V$ plot suggests that seifertite is more stable than cristobalite X-I at megabar pressures; however the difference is almost negligible.

**Towards the understanding of seifertite formation.** In agreement with previous studies[9,12,13,16,17], in experiments on powders of α-cristobalite loaded with and without the pressure-transmitting medium, we followed the sequence of transitions α→II→X-I by means of *in situ* Raman spectroscopy. The Raman spectra of powdered cristobalite X-I are weak and hardly detectable above 20 GPa (maximum pressure achieved was 50 GPa). However, upon decompression to ambient conditions, we were able to observe weak Raman scattering; the quality of the spectra improved significantly when the sample was extracted from the DAC and exposed directly to the laser beam. The so-obtained Raman spectrum of the sample is the same as reported earlier[12] for α-$PbO_2$-type silica (seifertite) and it is in good agreement with simulated spectra of seifertite[28] (Fig. 5).

This sample recovered after the DAC compression experiment up to 50 GPa (EXP 6, Supplementary Fig. 1) was mounted on a glass capillary and analysed using an in-house powder diffractometer. The two-dimensional image (Fig. 6) contains a halo at $d > 4.5$ Å, indicating large amounts of amorphous material. The poor crystallinity gives rise to broadened diffraction rings with weak intensities, such as no quantitative structural analysis was possible. However, the qualitative comparison of the position of the diffraction lines and their relative intensities show very good agreement with a product of a similar experiment on cristobalite cold-compression[12], as well as with seifertite formed at equilibrium (high PT) conditions[23]. All measured reflections correspond to seifertite, and we calculated the orthorhombic unit-cell parameters to be $a = 4.067(21)$ Å, $b = 5.090(38)$ Å, $c = 4.477(14)$ Å (Supplementary Table 5). Grain interaction within the powder samples obviously contributes to a non-hydrostatic environment even when neon was used, because it prevents the pressure-transmitting medium to be distributed among the fine-grained material. Therefore, the same product was formed regardless of the presence of neon.

Finally, this sample that started as powder of α-cristobalite and was identified upon recovery from 50 GPa, by means of Raman spectroscopy and X-ray diffraction as the same material on what was previously reported as seifertite, was analysed by TEM. Two different sample-preparation techniques were applied prior TEM,

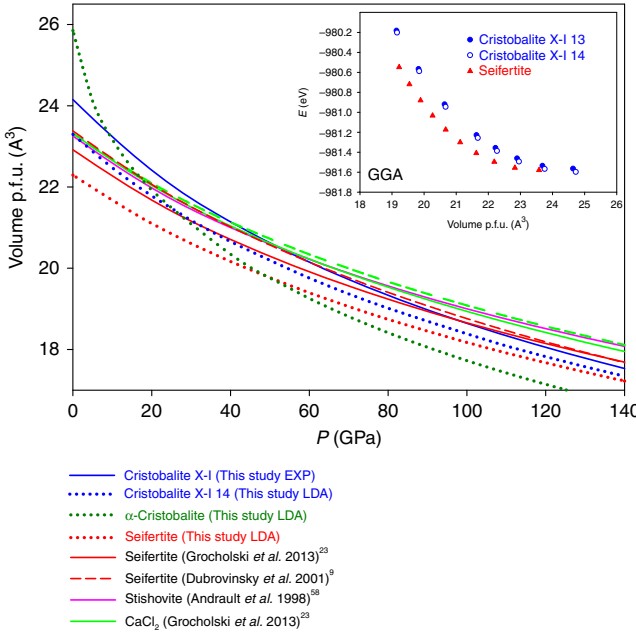

**Figure 4 | Molecular volume of various silica phases as a function of pressure.** They are calculated according to experimental equations of state given in refs 9,23,58 or implementing density-functional theory (DFT) with local-density approximation (LDA), starting from known structures[59,60]. Cristobalite X-I appears to be denser even than seifertite above about 100 GPa (experimental values). The theoretical structure of seifertite is about 3.5% denser than the experimental one, a characteristic limitation of the LDA functional. According to LDA, alpha-cristobalite would be the densest phase above ca. 50 GPa, but it has never been observed experimentally above 15 GPa. Generalized-Gradient approximation (GGA) calculations for seifertite and the two cristobalite X-I ordered phases are represented in the inset. See Supplementary Fig. 8 for $E$–$V$ curves of other phases.

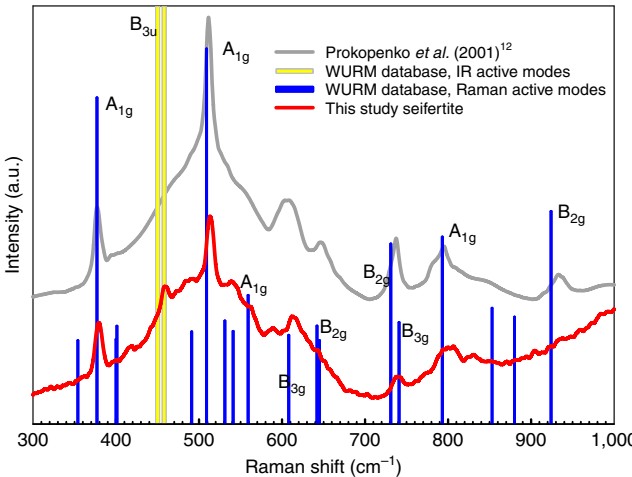

**Figure 5 | Raman spectrum of seifertite.** Sample from this study, quenched from 50 GPa (red line), collected outside the diamond-anvil cell (Dilor XY spectrometer, excitation wavelength 514 nm; power 0.7 W, acquisition time 120 s repeated three times). For comparison, we also plot previous experimental[12] (grey line) and computational[28] (labelled blue bars) Raman spectra of seifertite.

dispersion of powder on lacey carbon, and focused ion beam (FIB) foil extraction (see Sample recovery and preparation for TEM in Methods section). Both techniques showed similar

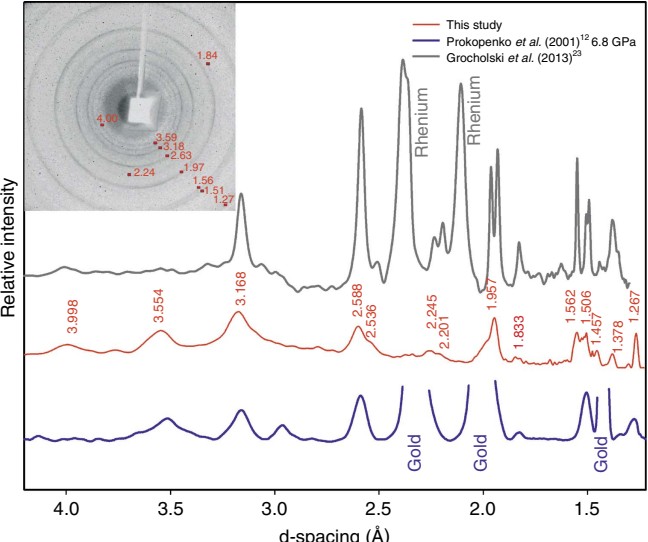

**Figure 6 | Powder-diffraction pattern of seifertite.** X-ray data collected at Rigaku diffractometer with MoKα radiation of the sample recovered after compression to about 50 GPa of the powder of α-cristobalite (red line in the middle). For comparison, we provide diffraction patterns of the material obtained by Prokopenko et al.[12] by compression–decompression of starting cristobalite sample to ∼50 GPa at ambient temperature and without pressure-transmitting medium and then quenched to ambient conditions α-PbO₂ structure (blue lower curve), as well as quenched seifertite synthesized at high PT conditions (grey upper curve, Grocholski et al.[23]). The similarity in the diffraction patterns is obvious.

results. The analysed material contained crystallites (ranging from ∼10 to 500 nm) surrounded by an amorphous matrix. The smallest crystallites were highly unstable under the beam and became amorphous instantly during data acquisition, yielding poor electron diffraction pattern. We performed the analyses by alternating between a direct SAED (large area, lower beam current, longer exposure) and a high resolution TEM mode (more selective, high beam current, shorter acquisition time), to collect diverse information of the material. Due to the destruction of the material, direct diffraction was rarely possible. From high resolution transmission electron micrographs (HRTEM) we derived FFT (Fast Fourier Transform) pattern. The $d$-spacings obtained by FFT can in some cases be interpreted as reflections of seifertite; however full indexing better agrees with the α-cristobalite unit cell rather than with the cell of seifertite (Fig. 7).

We suspect that electron irradiation might cause short-range transformation of unstable seifertite or seifertite-like phase to the more stable α-cristobalite. Similar back-transformation to α-cristobalite was observed when natural seifertite from shocked meteorites was analysed (personal communication El Goresy). Another possible explanation would be that by the time of TEM analyses, the entire material had transformed to α-cristobalite. Generally, small grains amorphize faster, which we interpret as caused by their larger surface area. The beam sensitivity is interpreted as a consequence of poor crystallinity. Following the interpretation by Biskupek and Kaiser[29], we find our maximal $\Delta d/d$ error of 6.3% for the (110_seifertite) acceptable (Fig. 7), given the small grain size, extraordinary instability of the sample under the beam as well as the fact that the sample represents not a compact but rather loose powder. However, the error was much smaller when indexing with cristobalite structure. We eliminated the possibility of ruby, graphite and diamond by EDS and indexing trials. The indexing of diffraction patterns seemed more efficient and less erroneous when direct SAED measurements

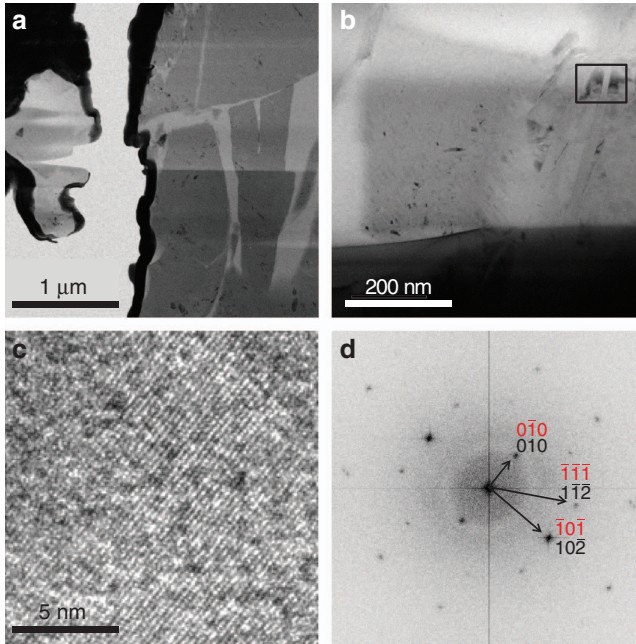

**Figure 7 | TEM analyses of the recovered samples.** Powder sample recovered from an experiment where starting α-cristobalite was cold-pressed to ca. 50 GPa and quenched to ambient conditions. (**a,b**) Bright field image showing isolated crystalline grains (dark grey) surrounded by amorphous matrix (light grey), with the rectangle in **b** marking the area selected for the HRTEM image shown in **c**. FFT (Fast Fourier Transform) diffraction pattern (**d**) derived from the area presented in **c**. The $d$-spacings obtained by FFT can be indexed as both α-cristobalite (black) and seifertite (red) in this particular direction. Due to high beam sensitivity, only one orientation of the specimen was acquired. The pattern seems to be more suitably indexed as α-cristobalite.

were performed. Due to the destruction of material, direct diffraction was rarely possible. In one such measurement, we were able to index only cristobalite and no seifertite, however, we cannot conclude based on this sole example that seifertite is completely absent from the sample. We mostly used FFT-derived diffraction from HRTEM images for indexing (Supplementary Table 5). In contrast, large quenched single-crystal α-cristobalite that back-transformed from cristobalite X-I was comparatively stable, easy to analyse and index (EXP 4, Supplementary Fig. 7).

Although the Raman spectra and powder X-ray diffraction pattern of the recovered phase are of insufficient quality to revel the structure of seifertite, they are in excellent agreement with other experimentally and theoretically obtained results on α-PbO$_2$ structured silica[12,23,28]. Therefore, we carefully chose not to name the quenched product seifertite, but rather a seifertite-like phase. Note, however, the exact seifertite structure has not been yet determined by means of single-crystal X-ray analysis.

Using powder of α-cristobalite as starting material in experiment, we lost the Raman signal from the sample above 20 GPa and we did not recover it up to the maximum compression (∼50 GPa) nor during slow decompression down to pressures very close to ambient. At room pressure we again could see a weak signal, which improved significantly by opening the DAC and exposing the sample to the laser beam directly (Fig. 5). Therefore, we cannot confirm that cristobalite X-I is the last phase occurring before formation of seifertite. However, as no changes in powder-diffraction pattern of this phase have ever been reported previously prior the transition to seifertite, it seems justified to assume that no other phases form in this sequence.

## Discussion

Most of cristobalite found in shocked materials is believed to form after pressure release due to high post-impact temperatures (for example, ref. 30). Here, we propose that in shocked meteorites[5–7] it can as well form as a product of cristobalite X-I back-transformation. High-pressure cristobalite X-I cannot be preserved at ambient conditions, although the reconstructive transition involving increase in coordination number of silicon from fourfold coordination in α- or cristobalite-II to the sixfold coordinated polymorph crsitobalite X-I does not require any thermal activation. This could explain why cristobalite can be found in impact-related rocks free of coesite and stishovite, but associated with other mineral assemblages formed at pressures exceeding 10 GPa.

Cristobalite was documented alongside and in close spatial relation with all the natural occurrences of the high pressure, post-stishovite silica polymorph seifertite[7,8] found in heavily shocked Martian or Lunar meteorites. Our experiments show that at non-hydrostatic environment and at ambient temperature conditions cristobalite follows α→II→X-I transformation path; at sufficiently high pressures (∼30–40 GPa)[9,14,18] it will eventually transform to quenchable seifertite or seifertite-like phase, bypassing the equilibrium formation of stishovite and CaCl$_2$-structured polymorph. A recent study has shown that applying moderate temperatures (700 K) can promote formation of seifertite at pressures as low as 12–13 GPa when starting from cristobalite X-I[18]. Our finding that 4-to-6 increase in coordination number of silicon in cristobalite X-I occurs as low as ∼10 GPa without requiring any heating, agrees well with the results by Kubo et al.[18]. Once the thermal energy is added, cristobalite X-I transforms to the more stable arrangement—seifertite, and not to the thermodynamically stable stishovite. This could also shade light onto coexistence of α-cristobalite, formed from quenched cristobalite X-I that is only stable at high pressures, with seifertite, which is quenchable, in meteorites in which peak pressures are estimated not to have exceeded 25–30 GPa. It is difficult to imagine that co-exiting, spatially related (sometimes as close as 200 nm[6–8]), particles of cristobalite and seifertite (and stishovite) were generated by differences in post-impact temperatures. Such temperature gradients/inhomogeneity is unrealistic[31], hardly achievable even by laser heating; at the same time huge variations in stress conditions in shock process are well-documented[32], and according to our observation could lead to very different transformation pathways of silica.

On the basis of our observation, we conclude that particularly in quasi-hydrostatic or non-hydrostatic conditions α-cristobalite may occur on the decompression path of cristobalite X-I. Thus, the presence of α-cristobalite does not exclude the possibility that a rock has experienced high-pressure conditions. Moreover, our results show that neither cristobalite nor seifertite should be considered as reliable tracers of the peak shock conditions. Seifertite occurs as small crystals in meteorites (at nanometer or micrometre scale[6–8]) and usually needs to be identified using TEM. Our results further suggest that seifertite may be misidentified as cristobalite, which can lead to erroneous interpretation of peak shock conditions.

Finally, the preservation of the cristobalite X-I structure under a broad pressure range and its survival at slightly elevated temperatures[18], suggests that this phase can be accommodated in quasi- or non-hydrostatic environments where temperature is insufficient to form coesite or stishovite, for example, in cold subducting slabs[33].

We do not agree with the recently proposed model[8] that densification of the fourfold coordinated cristobalite-II directly leads to an *hcp* arrangement of oxygen atoms representative of

seifertite. We observe intermediate sixfold cristobalite X-I in all our (quasi-hydrostatic and non-hydrostatic) experiments. Keeping in mind the similarity in structures of cristobalite X-I and seifertite, we conclude that if there were any phases intermediate between these two polymorphs, then they are formed by a simple rearrangement of the octahedra within the chains. Cristobalite X-I has so far the structure most similar to that of seifertite (containing kinked chains of octahedra) as deduced from precise single-crystal analysis and determination of exact atomic positions.

## Methods

**Starting material and high-pressure experiment set-up.** For the experiments, we used synthetic cristobalite as well as the natural samples kindly provided by Harvard Museum (Ellora Caves MGMH Cristobalite 97849). Composition of the starting material for synthesizing cristobalite was $SiO_2$ glass of very low impurity content[34]. Composition of natural cristobalite was measured using ICP-MS at Bayerisches Geoinstitut (BGI), and showed trace amounts of Al, Fe and Na of $\sim 1,000$ p.p.m. at maximum, comparable to that reported before[35]. Cristobalite was synthesized by heating the glass granules at temperatures of $\sim 1,650\,^{\circ}C$ for 3 h and then cooling down slowly over 15 h to $\sim 1,500\,^{\circ}C$ (that is, the temperature slightly above the cristobalite-tridymite transition), and then rapidly quenched. This procedure allowed the growth of cristobalite crystals up to $\sim 100\,\mu m$ in linear dimension; however it was not possible to avoid twinning which occurs on cooling due to the transition from the cubic $\beta$-cristobalite to the tetragonal $\alpha$-cristobalite. These crystals were of sufficient quality for Raman spectroscopy, but for single-crystal X-ray diffraction natural cristobalite had to be used.

High-pressure experiments were carried out using both single crystals and powders as starting material (Supplementary Fig. 1). All single-crystal experiments were conducted in quasi-hydrostatic conditions, using neon as pressure-transmitting medium, whereas powders were also studied in non-hydrostatic environment. We used piston-cylinder BX90 type DAC produced at BGI[36] with 250 μm culet size of the anvils. Samples were loaded together with a ruby sphere of $\sim 5\,\mu m$ in diameter into a cylindrical pressure chamber of 30–40 μm height and $\sim 125\,\mu m$ in diameter drilled in a pre-indented rhenium gasket. Neon, used as pressure-transmitting medium, was loaded into the DAC using the BGI gas loading system[37]. We chose neon as pressure-transmitting medium, because it had been shown that helium, providing slightly better hydrostaticity, could be easily incorporated in cavities of the cristobalite structure[38]. For Raman experiments, we used Ia type diamonds with low fluorescence. For *in situ* single-crystal X-ray diffraction we used Boehler-Almax diamond anvils[39] and apertures of 80°, to allow for large reciprocal space coverage.

**Raman spectroscopy.** Raman spectroscopy measurements were performed using a Dilor XY Raman spectrometer with $Ar^+$ ion laser (514.5 nm, Coherent Innova 300). The spectrometer is equipped with confocal optics, a CCD detector cooled by liquid nitrogen, and a double-stage spectrometer with a $1,800\,g\,mm^{-1}$ grating, resulting in spectral resolution of $1\,cm^{-1}$. The output power was varied according to the sample signal between 0.3 and 1.0 W. The spectrometer was calibrated using the silicon peak at $520\,cm^{-1}$. Raman spectra were collected between 100 and $1,200\,cm^{-1}$ in all DAC experiments at room temperature by pressurizing the starting material in steps of several GPa up to a maximum of $\sim 83$ GPa, and, in the same manner, also during decompression to ambient conditions. Data were typically collected for 120 s, using Labspec software. Peak positions in the Raman spectra were determined using Lorentzian peak fitting set-up in Igor Pro v. 6.22 software. The seifertite sample analysed by Raman spectroscopy remained stable after 120 s exposure to a 1.2 W direct $Ar^+$ laser beam. The natural samples of seifertite were reported to be very unstable under all kinds of electron or ion beams.

**In-house X-ray diffraction.** Single crystals of α-cristobalite were selected at the BGI based on their optical quality—size, transparency, crystal shape, surface appearance, and so on. Very small crystals (up to $\sim 30\,\mu m$ in linear dimension) were tested using a rotating anode high-brilliance Rigaku diffractometer with MoKα radiation ($\lambda = 0.7080\,\text{Å}$) operated at 60 kV and 50 mA, equipped with Osmic focusing X-ray optics and a Bruker Apex CCD detector. The same diffractometer was used to analyse the quenched powder from the EXP 6. The fine-grained powder was collected on a capillary and powder-diffraction data were collected for 3,600 s. To improve the quality of the data, we subtracted the pattern collected on an empty capillary for the same duration of time.

**Synchrotron X-ray diffraction.** The structural behaviour of cristobalite was investigated up to $\sim 83$ GPa at room temperature. High-pressure single-crystal X-ray diffraction experiments were performed at the beamlines ID09A and ID27 at ESRF (Grenoble, France) and at the Extreme Conditions Beamline (ECB, P02.2) at PETRA III in Hamburg. At each pressure point, wide-scan diffraction images were collected in the ω range of $\pm 25^{\circ}$ or $\pm 30^{\circ}$. In addition, 80 (or sometimes 160)

independent step-scan diffraction frames were collected with time exposure of 1 s per step in the same ω range. At ID09A, data were collected for the experiment BA-DAC1 up to the pressure of phase transition α-cristobalite to cristobalite –II ($\sim 4.5$ GPa) using 30 μm X-ray beam with a wavelength of 0.41298 Å and a MAR555 flat panel detector, located at the distance of 303.8 mm from the sample. At PETRA III, we continued the experiment on the same crystal up to $\sim 83$ GPa GPa, using $2 \times 4\,\mu m^2$ (FWHM) X-ray beam with a wavelength of 0.28962 Å. The detector used was a PerkinElmer XRD1621 flat panel, located at the distance of 398.8 mm. At ID27, we analysed new crystal loaded as BA-DAC2 at 14.1 GPa. A PerkinElmer flat panel detector was located at 387.0 mm. The wavelength of the used radiation was 0.3738 Å. Pressure was determined using the ruby pressure gauge[40] both before and after collection of X-ray. As well as using neon EOS reported by Fei et al.[41]. Single-crystal data have been reduced with the CrysAlis software package (Oxford Diffraction) and analysed with the SHELX97 (ref. 42) program package as implemented in the WingX software[43]. Isotropic structure refinements of intermediate polymorphs were performed based on $F^2$ starting from the atomic coordinates for α-cristobalite and cristobalite-II at ambient conditions, reported by Dera et al.[10]. The structure of cristobalite X-I was solved by direct method using SHELXS and refined by full matrix least squares in the isotropic approximation using SHELXL, both programs being implemented in SHELX97 software.

**Sample recovery and preparation for transmission electron microscopy.** A quenched single crystal (EXP 4) and a powder sample (EXP 6) were recovered after high-pressure experiments and prepared for transmission electron microscopy (TEM) analyses. The quenched crystal (EXP 4) was fixed by superglue to a glass, so that only a very thin layer of superglue was covering the surface ($<3\,\mu m$), and coated by a $\sim 15$ nm thin carbon layer. We prepared an electron-transparent foil out of glued crystal using a FIB-SEM instrument at the University of Vienna (see below). The agglomerate of several grains of the powder sample (EXP 6) was extracted directly from the gasket and first analysed by Powder X-ray Diffraction (PXRD). After that, we applied two different methods of the powder sample preparation for TEM analyses: (1) the powder sample was crushed between two glass slides and dissolved with a drop of ethanol. On this drop, we placed a copper grid with a lacey carbon film on which the powder remained fixed after the ethanol evaporated. However, because crushing the metastable material may cause spontaneous transformation, we also (2) prepared an electron-transparent foil out of the remnant material found inside the gasket using a FIB-SEM instrument at BGI (see below). The advantage of FIB compared to conventional Ar-ion thinning or crushing is that it result in very low heating[44] (few °C) and exerts no mechanical force on the sample. Electron-transparent lamellae were prepared following earlier descriptions[45,46].

**Focused ion beam sputtering.** We used two dual-beam instruments for the FIB preparation to produce $\sim 100$ nm thin, electron-transparent foils of the quenched material for TEM. FEI Quanta 3D FEG located at the Faculty of Geosciences, Geography and Astronomy at the University of Vienna was used for the quenched single crystal. Newly installed FEI Scios dual-beam instrument located at BGI, Bayreuth, was used for powder sample. Both instruments used focused $Ga^+$ ions accelerated by an accelerating voltage of 30 kV at ion beam currents in the range of 65 nA–30 pA for material sputtering. A platinum layer was deposited onto the sample surface for protection and enhancement of foil stability during FIB preparation. An Omniprobe 100.3 micromanipulator and an Easylift system with a tungsten needle were used in Quanta 3D and Scios, respectively, for *in situ* lift-out of the pre-thinned foil and transfer of the foil to a Cu-grid for final thinning.

**Transmission electron microscopy.** The FEI Titan G2 80–200 with ChemiSTEM technology at BGI was operated at an acceleration voltage of 200 keV. Even though the X-FEG allows obtaining very high brightness, we tuned the microscope to lower the current and minimize beam damage of the sensitive samples. The particles were either first imaged in bright field (BF) mode or in scanning transmission electron microscopy (STEM) mode. For the former, we either used a selected area aperture of 10 or 40 μm to obtain selected area diffraction pattern. The composition of the agglomerates was confirmed using EDX-ray spectroscopy either in STEM in combination with the high angle annular dark field , BF, dark field detector or in nano-diffraction. Some particles were imaged in high resolution mode and two-dimensional Fourier transformation patterns were calculated to obtain the d-spacings and angles between so-obtained diffraction spots. The chemistry of the grains was also cross-checked using EDX spectroscopy.

**Ab initio calculations.** We analyse the phonon softening from first-principles calculations. We determine the ground-state properties using standard DFT[47–49] with LDA in the ABINIT implementation, based on planewaves and pseudopotentials[50,51]. While LDA is notoriously wrong for quartz, for the silica phases stable at high pressure, it gives very good structural agreement with experimental data[52]. Starting from the crystal structure refined in this study, we determine the theoretical structure of cristobalite X-I at three different pressures: 10, 40 and 100 GPa. Then we compute the energy derivatives to build the dynamical matrices and the Raman tensors in the framework of the

density-functional perturbation theory[53–55]. We compare the theoretical Raman spectra with the measured ones at the same experimental density. This approach provides the most accurate theoretical description of spectroscopic data[28]. We employ a $4 \times 4 \times 4$ grid of special k points[56] to sample the electron density in the reciprocal space and a kinetic energy cutoff of 38 Hartrees (1 Hartree $= 27.2116$ eV). With these parameters the precision of the calculation is typically on the order of 0.001 Hartree in energy and better than 1 GPa in pressure. We computed the relative energy differences for selected tetrahedrally- and octahedrally-coordinated silica phases in LDA up to 140 GPa. We store all the Raman spectra computed under pressure on the WURM website (http://www.wurm.info and cristobalite specifically http://www.wurm.info/index.php?id=5). More details of the Raman calculations can be found in the original WURM paper[28]. However, due to the fact that LDA gives wrong tetrahedral-to-octahedral transition pressures for silica[52], we checked the relative energy differences between selected tetrahedrally-and octahedrally-coordinated silica phases using the GGA in the Perdew-Burke-Erzernhof formulation[57] for the exchange-correlation term.

**Data availability.** Raw data were partially generated at the large-scale facilities (Petra III, Germany and ESRF, France). Derived data supporting the findings of this study are available from A.Č. upon request. Cristobalite X-I structure in form of CIF file was deposited at CCDC under number 1491299. The computational data that support the findings of this study are available from www.wurm.info or from R.C. upon request. The data that support the findings of this study are available from the corresponding author upon reasonable request.

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

## Acknowledgements

A.Č. is truly grateful for the support from the University of Bayreuth Graduate School and for the Bavarian Gender Equality Grant, provided by the Bavarian State Government, which made this work possible. Hubert Schulze and Raphael Njul are gratefully acknowledged for the sample preparation, Andreas Audétat for helping us analyse the starting material using LA-ICP-MS and Tiziana Boffa-Ballaran for the help in data analysis and discussion. Portions of this research were carried out at the light source PETRA III at DESY, Helmholz group, as well as at the European Synchrotron Radiation Facility (ESRF). K.M. acknowledges the financial support through DFG grand Ma 6287/2 and Ma 6287/3. L.D. acknowledge support through DFG DU939/9, DU939/10, and BMBF grants. The authors acknowledge access to the Laboratory of SEM and FIB applications at the Faculty of Geosciences, Geography and Astronomy of the University of Vienna, Austria. The *ab initio* simulations were performed on the GENCI super-computers, under eDARI/CINES grants ×106368. Discussions and comments from P. Dera on our structural data are highly appreciated.

## Author contributions

L.D. and A.Č. planned and designed the research; A.Č. prepared all DACs; A.Č. performed Raman spectroscopy experiments; G.H., K.M. and A.Č. performed FIB preparation of TEM samples; K.M. and A.Č. collected and analysed TEM data; R.C. and E.Bo. performed theoretical calculations; A.Č, E.By. and L.D. collected synchrotron data with help and assistance of H.-P.L., M.H. and M.M.; E.By. and L.D. solved the structure of cristobalite X-I; A.Č. with contribution of L.D., R.C., and K.M. wrote the manuscript; all coauthors read and discussed the manuscript.

## Additional information

**Competing interests:** The authors declare no competing financial interests.

**How to cite this article:** Černok, A. *et al.* Compressional pathways of α-cristobalite, structure of cristobalite X-I, and towards the understanding of seifertite formation. *Nat. Commun.* **8,** 15647 doi: 10.1038/ncomms15647 (2017).

**Publisher's note**: 

