## [Peer Review File · Nature Communications]

Reviewers' Comments:

Reviewer #1 (Remarks to the Author)

In this manuscript Cernok et al. report on the possible transformation pathways of Cristobalite at high pressure, using a wealth of experimental approaches, including in situ Raman scattering at high pressure, X-ray diffraction (including Synchrotron), electron microscopy and ab initio calculations.

The topic is interesting and the findings reported in this paper have broad impact in Earth and Planetary science, high pressure physics, chemistry, and mineralogy.

The results reported in this work are new, coherent and discussed in an enthralling way: The authors manage to cast their findings in a broad context and put together an interesting story. The main findings are: (i) the characterization of cristobalite X-I, which has been so far a rather elusive polymorph of SiO₂, and its behaviour upon compression and decompression, including its surprising stability upon compression up to 80 GPa; (ii) proving the different effects of hydrostatic and non hydrostatic compression of cristobalite and identifying the pathway to Seifertite, which occur at lower pressure than expected.

For the importance of these findings and the broad range of methods employed to produce and validate them, this manuscript is definitely worth publishing in Nature Communications, provided that the authors take care of clarifying and improving a few issues:

1) Modeling is carried out by DFT and some details are provided in the methods section. One very important piece of information is missing: which density functional is used? In addition, the authors should mention how reliable their approach is for silica at high pressure.

2) Fig. 2 and 4 show the P-V equation of state of the newly characterized cristobalite X-I: it would be useful to show also the E(V) or H(V) theoretical curves, together with those of alpha-cristobalite and seifertite. It could be an inset of fig. 4 or a stand alone figure in SI.

3) The authors do not provide a satisfactory discussion of their results against former theoretical papers. In particular [L. Huang et al. Nature Mater. 5, 977 (2006)] should be mentioned, especially because the assignment of cristobalite X-I is completely different from what is proposed here, and it should be clarified.

4) Figure S4 is totally unclear to me. Even if the discussion in the text helps to understand what it should represent, it still remains very hard to read. A different representation, possibly showing also the octahedra, would help.

5) The similarity of the phonon dispersion relations in fig. S5 would be further supported if the integrated densities of vibrational states were shown. What about the phonons of the other configurations (a, d)?

Reviewer #2 (Remarks to the Author)

Report on the manuscript NCOMMS-16-15689 entitled "How cristobalite transforms to seifertite" submitted by A. Cernok, K. Marquardt, R. Caracas, et al.

The authors claimed in the submitted manuscript to provide evidence for the formation of seifertite in shocked meteorites via a non-equilibrium and strongly non-hydrostatic compression of alpha-cristobalite -> cristobalite II -> cristobalite X-I and finally to seifertite by cold-compression DAC experiments starting either with single crystals or powders. The phase transitions were studied by in-situ and partly ex-situ Raman spectroscopy, single crystal / powder X-ray diffraction and TEM. The crystal structure of the intermediate phase cristobalite X-I has been determined for the first

time by single crystal X-ray diffraction.

The study appears technically sound, is presented in clear and fluent English. It provides several relevant new issues for the interested community, which should be published, like the structure of cristobalite X-I or the strong stress sensitivity of the transformation, due to which any study distinguishes slightly from former reported ones in their transition pressures. Compared to the accurately studied transition up to cristobalite X-I and its structural presentation, unfortunately, the presented observations regarding the final formation of seifertite are rather poor. Presented conclusions of this transition appear exaggerated and to my opinion not unambiguously true, they remind more to model than a real observation. The presented path is neither original nor new, anyway. Therefore I suggest the manuscript would benefit from a revision pointing out more clearly the new aspects of the work and a more clear separation from already in the literature described trends and speculations.

Comments in detail:

Formation of seifertite: Whereas the transition up to cristobalite X-I could be observed in-situ by Raman spectroscopy and X-ray diffraction, the seifertite formation has only been observed ex-situ, because in-situ signals got too weak and broad. This is a known problem, but the presented ex-situ observations provide weak evidence for seifertite. The Raman spectra is identified by comparison with another quenched and similar spectra of ref [10], which authors called it cristobalite IV and suggest similarities with the alpha-PbO₂ type, but e.g. provide lattice parameters, which are different from those known for seifertite today and the speculated lattice distortions are not known from natural seifertite or most other studies. Their spectra recorded at high pressures looks significantly different, and after pressure release the broad band about 500 cm⁻¹ is inevitably related to the decomposition of large parts of the structure and only a few bands of the calculated bands persist. The laser-beam sensitivity of seifertite is known (e.g. [4]) and the author's just mention that their sample is unexpectedly stable without discussing why (probably it was not). Since Raman spectroscopy is a very local probe, the data show evidence for some phase showing amorphous and crystalline fragments, which show similar structural units (e.g. connectivity of Si-octahedra) to seifertite, but must not correspond to remaining part of real crystalline seifertite as it occurs in meteorites. Presented d-spacings from X-ray diffraction show beside some values known only from seifertite, several others, which do not correspond to well-crystalline seifertite and some - even strong - seifertite reflections are missing, which could not be due to orientational statistics, because in Fig 6 authors present homogeneous diffraction rings. Again, this suggests a decomposition product of something, which is not unambiguously seifertite, only something structurally related. Finally the authors admit, that crystals investigated by TEM are more likely indexed by alpha-cristobalite than seifertite, which is fully correct, because expecting for one phase larger errors in d-spacings than for others observed in the same TEM session is technically impossible. Why not have a look and present a "true seifertite crystal", if present, all the rest of the study is investigated in so much detail. Even the twins in Fig SM7 remind more to cristobalite (the black areas in BF are due to strong diffraction contrast, they do not indicate boundaries of twin lamellae and part of the surrounding material is not amorphous, just far out of diffraction condition). Interestingly, the authors called the evidence for that transition inconsistently in the abstract "eventually" (L26) or seifertite-like (L179), whereas otherwise they state "very good agreement" for Raman (L173) or "all measured reflections correspond to seifertite" (L175).

Path of phase transitions: Several times the authors point out that the observed phase transition occur even without thermal activation (e.g. L200, L211). This implies that the energetic barrier of the phase transition is found to be so low, that cold compression shows the energetically most favourite path. However, this would only be true for thermodynamically equilibrated transitions (ignoring entropy). For seifertite, all the recent paper point out since years, that it must have formed by non-equilibrium processes and hence it is no surprise that the transition path is bypassing thermodynamically stable phases (e.g. L 207). For non-equilibrium processes, high temperatures processes, including reconnecting bonds or short diffusion processes, rival with the observed processes of cold deformation. Hence temperature might significantly modify results of cold compression compared to Hugoniot-heating during an impact. This is the same problem for

the study of [11], which even admit, that heating after compression will enhance a seifertite formation. The suggested path of α -crist. \rightarrow crist.II \rightarrow crist. X-I \rightarrow seifertite is more a formation model than a proof observation. Even the authors cited all the previous studies resulting in a similar transition path correctly; the reader got the delusive impression that the confirmation of this path is a central result of the manuscript L71, L103. But in fact this transition path is known for years from several studies using cold compression experiments. For technical reasons, the authors can even not exclude the possibility of further intermediate 'seifertite like' phases (mentioned correctly, L235-237) or even such an end product (see above). Hence I suggest the title and parts of the introduction and discussion is a bit an overstatement.

Several times the authors cite the literature correctly but tend to report only half-truth in the manuscript leading to a slight overselling of their statements. E.g. they report on the "curious paragenesis" of seifertite and cristobalite (L18, L43, L47, L203) and that "no consensus" has emerged from "discrepant results" in the literature (L50/51). This is not completely true, since a true "paragenesis" is only a valid expression for equilibrated phase assemblages and it is known since more than ten years that this is not valid for seifertite. In detail only one of the cited work [5] reports from coexisting seifertite + cristobalite in lunar meteorites and a further [4] is suggesting an intergrown of these phases due to back-transformations. The vast majority of studies describe, that several silica high pressure polymorphs occurring in one meteorite are separated into different phase assemblages. The related discrepancies in pressure are known and are interpreted as a hint for a metastable seifertite formation, so most studies not assume extremely high shock pressures or certain peak shock pressures due to the formation of seifertite since years.

Structure of cristobalite V-I: I think the presentation of a crystal structure of cristobalite X-I is one of the outstanding results of that paper, because several previous studies failed due to the bad quality of powder pattern, but knowledge of the structure is necessary e.g. for ab-initio calculation but difficult to obtain. Even it is true that this structure has never been solved by single crystal diffraction before, the study of Huang et al. (2006) Nat.Mater.5, 977 present a significantly deviating structure of cristobalite X-I with 4-fold coordinated silicon by ab-initio calculation, which was later used for calculation e.g. of [12] and show congruent powder pattern compared to experimentally measured ones. The R-factors of this study are still relatively high (likely due to the difficult measurements of the light elements in the phase), but I'm wondering why clearly different structural solutions result in such similar looking powder pattern and I guess a self-critical discussion of this issue by the authors would help the readers to relate the presented solutions.

Effect of non-hydrostaticity: Different transformation pressures in single crystals are related to the axial stress which larger crystals accumulated by touching the anvils after closing the DAC. I'm wondering why this cause only a transition into a single crystal of cristobalite X-I at lower pressures and will not crack the single crystal or even grind it to powder (which of cause will transform earlier). The axial stress per definition depends on its crystallographic direction, so along which direction are the crystals aligned in the DAC and is there an difference between both crystals or could that be a test of the hypothesis if another crystal would be realigned by 90°. A different axial stress direction is proposed theoretically for the transformation to seifertite. A potential reason why the seifertite transition is not observed for single crystals?

Reviewer #3 (Remarks to the Author)

The review of Černok et al.

The authors conducted ultra-high pressure synthetic experiments of α -cristobalite under ambient temperature conditions using a diamond anvil cell (DAC) to test its phase transition process. They used single crystal α -cristobalite and powdered α -cristobalite as a starting material. The DAC experiments were performed under hydrostatic, quasi-hydrostatic and non-hydrostatic conditions. Under hydrostatic condition α -cristobalite remained its original structure up to 15 GPa. On the

other hand, under non-hydrostatic condition, α -cristobalite transforms into cristobalite X-I subsequent to cristobalite II around 10 GPa. The cristobalite X-I was observed up to around 80 GPa. The Raman spectra of powdered cristobalite X-I under non-hydrostatic condition are weak and hardly detectable above 20 GPa. The powder sample was recovered after decompression. The recovered sample was identified as seifertite by XRD analysis. Their experiments imply that the phase transitions of α -cristobalite depend on the degree of hydrostatic. It is surprising that the phase transitions proceed under room temperature condition. The authors also propose based on their DAC experiments that most of α -cristobalite in shocked meteorites is the back-transformation products of cristobalite X-I. The authors find new curious behaviors of silica phase. However, several portions in present Ms are confusing and misleading. Major portions should be addressed are as follows.

1) The authors mention the coexistence of α -cristobalite and high-pressure polymorphs in shocked meteorites and try to adopt their results to estimate shock pressure conditions. Please note that temperature distribution in the shocked meteorites is heterogeneous. Only shock-induced melting portions (shock-melt vein and melt-pocket) in the shocked meteorites experienced very high-temperature conditions beyond the liquidus temperature of the meteorites besides high-pressure conditions. High-pressure polymorphs occur only in and around the shock-induced melting portions. Even though α -cristobalite occurs in the host-rock of a shocked meteorite without coesite or stishovite and a high-pressure polymorph occurs in the shock-induced melting portions of same shocked meteorite, it is not unusual. If the authors want to compare the phase transitions of α -cristobalite in shocked meteorites and your results obtained through DAC experiments, they should consider the heterogeneous temperature distribution and the occurrences of α -cristobalite and high-pressure polymorphs.

2) The authors compare the phase transitions of α -cristobalite under hydrostatic, quasi-hydrostatic and non-hydrostatic conditions. The α -cristobalite under hydrostatic condition did not transform into any other phase over 15 GPa? In present MS, the descriptions focus just on under quasi-hydrostatic and non-hydrostatic conditions. The authors should clearly describe also the phase transitions of cristobalite under hydrostatic condition if you want to compare them.

3) How do you define the hydrostatic, quasi-hydrostatic and non-hydrostatic conditions in your experiments? Please explain it.

4) In line 77-78, the authors describe that "All single crystal experiments were conducted in quasi-hydrostatic conditions". On the other hand, in line 91 they say that "...and thus experience hydrostatic environment". This is confusing. Please clarify it.

5) Line 196-197: This is not widely accepted in shocked meteorites. Please note that the reference you citing is the case of a shocked rock around a crater. In general, cristobalite in meteorites is regarded as a magmatic product.

6) Line 197-168: Assuming that α -cristobalite is the back-transformation product of cristobalite X-I, what is an original silica phase before an impact in a meteorite? If the original silica phase is quartz or tridymite, they also have same phase transition path?

7) Line 201-202: Again, heterogeneous temperature distribution in shocked meteorites should be considered.

8) Line 212-213: Based on Kubo et al. (2015), it depends on the duration of P and T.

9) Line 228-230: Terrestrial silica phase is mainly quartz. Quartz behaves like α -cristobalite? Your explanation is needed here.

Others

The authors should distinguish your opinions (or interpretations) from results if you prefer to use both "results" and "discussion" sections. For example, you mention that "We suspect that electron irradiation might cause short-range transformation of unstable seifertite to the more stable α -cristobalite." in results section (line 189-191). This should be included in your discussion section.

Reviewer #1

“In this manuscript Cernok et al. report on the possible transformation pathways of Cristobalite at high pressure, using a wealth of experimental approaches, including in situ Raman scattering at high pressure, X-ray diffraction (including Synchrotron), electron microscopy and ab initio calculations.

The topic is interesting and the findings reported in this paper have broad impact in Earth and Planetary science, high pressure physics, chemistry, and mineralogy.

The results reported in this work are new, coherent and discussed in an enthralling way: The authors manage to cast their findings in a broad context and put together an interesting story.

The main findings are: (i) the characterization of cristobalite X-I, which has been so far a rather elusive polymorph of SiO₂, and its behaviour upon compression and decompression, including its surprising stability upon compression up to 80 GPa; (ii) proving the different effects of hydrostatic and non hydrostatic compression of cristobalite and identifying the pathway to Seifertite, which occur at lower pressure than expected.

We highly appreciate Reviewer's #1 support and recognition of the importance of our work.

For the importance of these findings and the broad range of methods employed to produce and validate them, this manuscript is definitely worth publishing in Nature Communications, provided that the authors take care of clarifying and improving a few issues:”

Q1) “Modeling is carried out by DFT and some details are provided in the methods section. One very important piece of information is missing: which density functional is used? In addition, the authors should mention how reliable their approach is for silica at high pressure.”

A1: We used the local-density approximation, and specified it in the revised form of the manuscript. We also comment on its reliability: ” While LDA is notoriously wrong for quartz, for the silica phases stable at high pressure, it gives very good structural agreement with experimental data³⁸.”

With ref. 40: Th Demuth, Y Jeanvoine, J Hafner, & J G Angyan Polymorphism in silica studied in the local density and generalized-gradient approximations, J. Phys.: Condens. Matter 11, 3833–3874 (1999).

We also specify more details about the computed Raman spectra:

“We compare the theoretical Raman spectra with the measured ones at the same experimental density. This approach provides the most accurate theoretical description of spectroscopic data²⁶”

Q2) “Fig. 2 and 4 show the P-V equation of state of the newly characterized cristobalite X-I: it would be useful to show also the E(V) or H(V) theoretical curves, together with those of alpha-cristobalite and seifertite. It could be an inset of fig. 4 or a stand alone figure in SI.”

A2: We have decided to compute the theoretical P-V values for both alpha-cristobalite and for seifertite, and to present them along with other molar volumes in the revised Fig. 4. That way we can compare all theoretical and experimental values.

“Q3) The authors do not provide a satisfactory discussion of their results against former theoretical papers. In particular [L. Huang et al. Nature Mater. 5, 977 (2006)] should be mentioned, especially because the assignment of cristobalite X-I is completely different from what is proposed here, and it should be clarified.”

A3) We appreciate Reviewer’s #1 comment, and we are very well aware of the work by Huang et al. Following Reviewer’s #1 advice, we add references on the previous work, as well as several lines about previous studies on silica under pressure:

“Earlier theoretical simulations^{13,14,21,22} did not predict the structure of cristobalite X-I observed in our study. Molecular dynamics simulations on cristobalite supercells¹⁴ obtained a compact structure with SiO₄ tetrahedra at high pressures, which eventually transformed into stishovite under further compression. Metadynamics simulations starting with quartz yield metastable kinked octahedral chains under compression¹³. Various other chain-like configurations have been proposed previously for the high-pressure forms of silica^{21,22}... “ at the beginning of the 4th paragraph of the Crsitobalite X-I section.

We state clearly in the text that earlier theoretical simulations did not predict the structure of Cristobalite X-I as we experimentally found it. However, we believe that a detailed discussions on why is it so would go well beyond the framework of the current paper. We will certainly follow suggestion made by Reviewer #1 and will consider this problem in future. We replace Ref 14 with Huang et al. 10.1038/nmat1760.

Q4) “Figure S4 is totally unclear to me. Even if the discussion in the text helps to understand what it should represent, it still remains very hard to read. A different representation, possibly showing also the octahedra, would help.”

A4: We have modified the figure SM4. We decided to represent the Si-O bonds rather than the SiO₆ polyhedra, in order to complement with the Fig. 2 in the main text. We also changed the caption, which we agree was not very clear. Following your advice, we decided to represent theoretical 4x4 arrangement with octahedra together with the experimental structure in the revised Fig. 2 in the main text.

Q5) “The similarity of the phonon dispersion relations in fig. S5 would be further supported if the integrated densities of vibrational states were shown. What about the phonons of the other configurations (a, d)?”

A5: We have modified figure SM5 to show also the density of states for the two configurations. As the Raman spectra show, we expect that the phonons for the mixed configurations to show no more complexity or particular features than the ones of the individual configurations.

Reviewer #2

“The authors claimed in the submitted manuscript to provide evidence for the formation of seifertite in shocked meteorites via a non-equilibrium and strongly non-hydrostatic compression of alpha-cristobalite -> cristobalite II -> cristobalite X-I and finally to seifertite by cold-compression DAC experiments starting either with single crystals or powders. The phase transitions were studied by in-situ and partly ex-situ Raman spectroscopy, single crystal / powder X-ray diffraction and TEM. The crystal structure of the intermediate phase cristobalite X-I has been determined for the first time by single crystal X-ray diffraction.

The study appears technically sound, is presented in clear and fluent English. It provides several relevant new issues for the interested community, which should be published, like the structure of cristobalite X-I or the strong stress sensitivity of the transformation, due to which any study distinguishes slightly from former reported ones in their transition pressures. The study appears technically sound, is presented in clear and fluent English. It provides several relevant new issues for the interested community, which should be published, like the structure of cristobalite X-I or the strong stress sensitivity of the transformation, due to which any study distinguishes slightly from former reported ones in their transition pressures.”

We are glad that Reviewer #2 has a positive view on our work.

Q1) “Compared to the accurately studied transition up to cristobalite X-I and its structural presentation, unfortunately, the presented observations regarding the final formation of seifertite are rather poor. Presented conclusions of this transition appear exaggerated and to my opinion not unambiguously true, they remind more to model than a real observation. The presented path is neither original nor new, anyway. Therefore I suggest the manuscript would benefit from a revision pointing out more clearly the new aspects of the work and a more clear separation from already in the literature described trends and speculations.

Formation of seifertite & Path of phase transitions:: Whereas the transition up to cristobalite X-I could be observed in-situ by Raman spectroscopy and X-ray diffraction, the seifertite formation has only been observed ex-situ, because in-situ signals got too weak and broad. This is a known problem, but the presented ex-situ observations provide weak evidence for seifertite... (three paragraphs not copied here) **Hence I suggest the title and parts of the introduction and discussion is a bit an overstatement.”**

A1) We are thankful for directions provided by Reviewer #2 and we hope that she/he finds the revised manuscript more suitable. We tried to address the question below in as much detail as we could. We fully agree with the Reviewer #2 that the formation of seifertite on compression of cristobalite was described in literature at least 15 years ago (in some cases seifertite was not actually recognized, and some of the co-authors of this paper were among those who realized this). Moreover, we provide in the manuscript quite a comprehensive list of references where this phenomenon was reported (Refs. 6,7,11,18,22,24). Thus detection or structural characterization of seifertite/alpha-PbO₂ structured silica was not a goal of current study; we merely confirm that we observed the same phase (and with comparable quality of experimental evidences) with what was determined as seifertite/alpha-PbO₂ structured silica in previous publications. Moreover, poor crystallinity of seifertite obtained at room temperature excludes the possibility of any detailed structural studies, but delivers the same conclusion on nature of the material as it was known from previous publications. Our new results are **(a)** a demonstration that a phase (cristobalite X-I) which

precedes the formation of seifertite at relatively low pressures (just at about 10 GPa) also has a structure based on SiO_6 octahedra, most similar to that of seifertite ever to be studied as single crystal (b) cristobalite found in mineralogical assemblages that faced high shocked pressure may be as well a product of decompression of cristobalite X-I (because we found that in quasi-hydrostatic conditions cristobalite X-I may survive at pressures over 60 GPa and thus mask real peak pressure of the impact process), and (c) the transformational path of cristobalite-derived polymorphs strongly depend on hydrostaticity of the environment and thus pressure range at which seifertite may occur is extremely large. We did our best to focus manuscript on our own new results and conclusions. Following the Reviewer's #2 suggestion to better emphasize the originality of our work, we modified the title of manuscript, the abstract, and some parts of the text. Thus, we address the findings related to high-pressure behavior of cristobalite rather than on those on seifertite formation. Due to the nature of the material recovered after non-hydrostatic experiments, we unfortunately cannot provide more detailed structural analyses of it. We hope the Reviewer #2 will appreciate the changes in the text.

Q2) “Several times the authors cite the literature correctly but tend to report only half-truth in the manuscript leading to a slight overselling of their statements. E.g. they report on the “curios paragenesis” of seifertite and cristobalite (L18, L43, L47, L203) and that “no consense” has emerged from “discrepant results” in the literature (L50/51). This is not completely true, since a true “paragenesis” is only a valid expression for equilibrated phase assemblages and it is known since more than ten years that this is not valid for seifertite. In detail only one of the cited work [5] reports from coexisting seifertite + cristobalite in lunar meteorites and a further [4] is suggesting an intergrown of these phases due to back-transformations. The vast majority of studies describe, that several silica high pressure polymorphs occurring in one meteorite are separated into different phase assemblages. The related discrepancies in pressure are known and are interpreted as a hint for a metastable seifertite formation, so most studies not assume extremely high shock pressures or certain peak shock pressures due to the formation of seifertite since years.”

A2) We believe that there may be a misunderstanding in terminology: there are (at least) three different definitions of term “paragenesis” – (a) repeatedly occurring in close spatial relations combination of minerals (most general definition we refer to), (b) mineral assemblages of sequentially formed species (definition often used in literature dealing with ore formations), and (c) equilibrated phase assemblages (as Reviewer #2 used it; however, to our understanding it would be rather difficult to apply such definition to any naturally occurring shocked material. As we noticed that this word has caused some misunderstanding, we exclude the word “paragenesis” from the manuscript in order to avoid any further terminological disputes and uncertainties.

Moreover, we do not really agree that “coexisting seifertite + cristobalite” were only reported in Refs 4 and 5. In fact even in first study conducted by Prof. T. Sharp, TEM analyses of “the post-stishovite silica polymorph” (later identified as a-PbO₂ structured silica), cristobalite was observed and was interpreted as a result of electron-beam damage of high-pressure phase. We had intensive discussion with Prof. A. El Goresy (who actually discovered “seifertite” and later named it), and he pointed out to a range of confusing and controversial literature reports on peak pressure assessment in meteorites/rocks containing seifertite. As Prof. A. El Goresy pointed out, the issue on how seifertite forms, particularly in the context of itsco-existence with low-pressure cristobalite, is one of the most important remaining puzzles in peak shock pressure estimations. We do not intend to enhance discussion of this subject in the paper because it

will drive us out of the topic and anyhow we will not be able to describe problem better than an expert himself, Prof. A. El Goresy; he is currently preparing a review article on this topic.

It is correct that words “metastable formation” could characterize reason for seifertite appearance, but they do not explain anything. Earlier experimental works reported direct transformation pressure from cristobalite in to seifertite above about 30 GPa, too high for other minerals observed to coexist; very nice paper by Kubo et al. (Ref. 11) demonstrate that seifertite may occur at pressures as low as 11 GPa and elevated temperatures, but authors relate appearance of seifertite to kinetic factors (and thus to the size of impactor body), and not to degree of hydrostaticity (which we demonstrate could play crucial role in transformation path). Moreover, Kubo et al. did not know that cristobalite X-I is another silica phase build of SiO_6 octahedra, that it can persist to very high pressures (over 60 GPa), and it can back-transform on decompression to cristobalite thus explaining why different silica polymorphs may be found together.

Q3) “Structure of cristobalite V-I: I think the presentation of a crystal structure of cristobalite X-I is one of the outstanding results of that paper, because several previous studies failed due to the bad quality of powder pattern, but knowledge of the structure is necessary e.g. for ab-initio calculation but difficult to obtain. Even it is true that this structure has never been solved by single crystal diffraction before, the study of Huang et al. (2006) Nat.Mater.5, 977 present a significantly deviating structure of cristobalite X-I with 4-fold coordinated silicon by ab-initio calculation, which was later used for calculation e.g. of [12] and show congruent powder pattern compared to experimentally measured ones. The R-factors of this study are still relatively high (likely due to the difficult measurements of the light elements in the phase), but I’m wondering why clearly different structural solutions result in such similar looking powder pattern and I guess a self-critical discussion of this issue by the authors would help the readers to relate the presented solutions.”

A3) We would like to avoid transforming our paper in to a technical comment on previous publications, and thus we prefer not to argue too much on predictions by Huang et al. or Donadio et al. A closer look at Fig. 2a in Huang et al. (now Ref. 14) clearly shows large disagreement between experimental data (even if only in regard to the peaks positions) and calculations.

We would like Reviewer #2 to know that we provided our data to Prof. P. Dera from the University of Hawaii, (a well-known expert in high pressure crystallography and author on previous publications on high-pressure silica phases) and he independently confirmed our structural results.

We have added several lines about previous studies on silica under pressure to address disagreement with previous theoretical studies:

“Earlier theoretical simulations^{13,14,21,22} did not predict the structure of cristobalite X-I observed in our study. Molecular dynamics simulations on cristobalite supercells¹⁴ obtained a compact structure with SiO_4 tetrahedra at high pressures, which eventually transformed into stishovite under further compression. Metadynamics simulations starting with quartz yield metastable kinked octahedral chains under compression¹³. Various other chain-like configurations have been proposed previously for the high-pressure forms of silica^{21,22}... “ at the begining of the 4th paragraph of the Crsitobalite X-I section.

Q4) “Effect of non-hydrostaticity: Different transformation pressures in single crystals are related to the axial stress which larger crystals accumulated by touching the anvils after closing the DAC. I’m

wondering why this cause only a transition into a single crystal of cristobalite X-I at lower pressures and will not crack the single crystal or even grind it to powder (which of cause will transform earlier). The axial stress per definition depends on its crystallographic direction, so along which direction are the crystals aligned in the DAC and is there an difference between both crystals or could that be a test of the hypothesis if another crystal would be realigned by 90°. A different axial stress direction is proposed theoretically for the transformation to seifertite. A potential reason why the seifertite transition is not observed for single crystals?"

A4) We appreciate Reviewer's #2 advices to study in details effect of orientation on transformational path in cristobalite, and hope to test it in more detail in the near future (this obviously requires preparation of several crystals with different orientation but identical thickness, to load them in the same cell and with the same pressure medium, etc. Due to a technical complexity of such approach, reviewer will understand that this needs to be a separate project). As we describe in the paper, we performed several independent experiments on cristobalite of different origins, loaded in different cells in different orientations. In addition to that, just like previous studies, we also observe that the powder sample transformed to cristobalite-II and X-I at comparable pressures, which indicates that random orientations did not contribute to any change of transformation pressure regarding these two polymorphs. At the moment we do not have any compiling evidences that orientation of the crystal with respect to stress tensor plays any role, but as we mention above, we do not have enough arguments to exclude this.

In experiments we performed with deliberately bridged between anvils crystals thickness of the sample was carefully selected to avoid it crashing or smashing in to powder.

Reviewer #3

“The authors conducted ultra-high pressure synthetic experiments of α -cristobalite under ambient temperature conditions using a diamond anvil cell (DAC) to test its phase transition process. They used single crystal α -cristobalite and powdered α -cristobalite as a starting material. The DAC experiments were performed under hydrostatic, quasi-hydrostatic and non-hydrostatic conditions. Under hydrostatic condition α -cristobalite remained its original structure up to 15 GPa. On the other hand, under non-hydrostatic condition, α -cristobalite transforms into cristobalite X-I subsequent to cristobalite II around 10 GPa. The cristobalite X-I was observed up to around 80 GPa. The Raman spectra of powdered cristobalite X-I under non-hydrostatic condition are weak and hardly detectable above 20 GPa. The powder sample was recovered after decompression. The recovered sample was identified as seifertite by XRD analysis. Their experiments imply that the phase transitions of α -cristobalite depend on the degree of hydrostatic. It is surprising that the phase transitions proceed under room temperature condition. The authors also propose based on their DAC experiments that most of α -cristobalite in shocked meteorites is the back-transformation products of cristobalite X-I. The authors find new curious behaviors of silica phase.”

We appreciate Reviewer’s #3 careful summary of our main results.

Q1) “The authors mention the coexistence of α -cristobalite and high-pressure polymorphs in shocked meteorites and try to adopt their results to estimate shock pressure conditions. Please note that temperature distribution in the shocked meteorites is heterogeneous. Only shock-induced melting portions (shock-melt vein and melt-pocket) in the shocked meteorites experienced very high-temperature conditions beyond the liquidus temperature of the meteorites besides high-pressure conditions. High-pressure polymorphs occur only in and around the shock-induced melting portions. Even though α -cristobalite occurs in the host-rock of a shocked meteorite without coesite or stishovite and a high-pressure polymorph occurs in the shock-induced melting portions of same shocked meteorite, it is not unusual. If the authors want to compare the phase transitions of α -cristobalite in shocked meteorites and your results obtained through DAC experiments, they should consider the heterogeneous temperature distribution and the occurrences of α -cristobalite and high-pressure polymorphs.”

A1: We are fully agreed with Reviewer #3. In the paper we state: “Most of cristobalite found in shocked materials is believed to form after pressure release due to high post-impact temperatures^{e.g.27}. Here we propose that in shocked meteorites^{4,5} it can as well form as a product of cristobalite X-I back-transformation.” Moreover, in the revised manuscript we on several occasions underline role of temperature in shocked-induced processes. We add to discussion: It is difficult to imagine that co-existing, spatially related (sometimes as close as 200 nm⁴⁻⁶), particles of cristobalite and seifertite (and stishovite) were generated by differences in post-impact temperatures. Such temperature gradients/inhomogeneity is unrealistic²⁹, hardly achievable even by laser heating; at the same time huge variations in stress conditions in shock process are well-documented, and according to our observation could lead to very different pathways of silica transformation pathways.

Q2 and 3) “The authors compare the phase transitions of α -cristobalite under hydrostatic, quasi-hydrostatic and non-hydrostatic conditions. The α -cristobalite under hydrostatic condition did not transform into any other phase over 15 GPa? In present MS, the descriptions focus just on under quasi-hydrostatic and non-hydrostatic conditions. The authors should clearly describe also the phase transitions of cristobalite under hydrostatic condition if you want to compare them.

How do you define the hydrostatic, quasi-hydrostatic and non-hydrostatic conditions in your experiments? Please explain it.”

A2 and 3: In the revised version of the manuscript we explain that under hydrostatic conditions we understand those provided by liquid pressure transmitting medium (for example, methanol-ethanol up to about 10 GPa), quasi-hydrostatic – in solidified but very soft pressure transmitting medium (like in Ne up to about 50 GPa, see, for example, Takemura et al. *Journal of Physics: Conference Series* (2010) 215, 012017-2;), and non-hydrostatic – when material compressed in solid pressure transmitting medium or without any medium (or sample is bridge between anvils, or experience intergranular stresses like in powders). We indeed do not observe phase transitions in α -cristobalite up to 15 GPa (Fig. 1b) compressed in (almost) hydrostatic conditions (small crystal totally surrounded by Ne which at this pressure may be considered as hydrostatic medium because it is not even possible to measure stress in it, see ref on Takemura et al. above).

Q4 “In line 77-78, the authors describe that "All single crystal experiments were conducted in quasi-hydrostatic conditions". On the other hand, in line 91 they say that "...and thus experience hydrostatic environment". This is confusing. Please clarify it.”

A4: We modified statement as following: “crystal (~10 μm thickness, “C3”) that does not bridge between the anvils and fully surrounded by very soft neon, and thus experience (almost) hydrostatic environment...”; see also discussion above.

Q5 “Line 196-197: This is not widely accepted in shocked meteorites. Please note that the reference you citing is the case of a shocked rock around a crater. In general, cristobalite in meteorites is regarded as a magmatic product.”

A5: We agree with Reviewer #3 and it follows from our formulation (“shocked materials”). We further underline that our proposal is related particularly to shocked meteorites by adding refs #4 and 5 in the corresponding sentence.

Q6 “Line 197-168: Assuming that α -cristobalite is the back-transformation product of cristobalite X-I, what is an original silica phase before an impact in a meteorite? If the original silica phase is quartz or tridymite, they also have same phase transition path?”

A6: We are fully agreed with Reviewer #3 that it may be very interesting option to figure out what was starting silica phase because transformation path is highly starting-material dependent. At the moment we can for sure say that quartz does not have the same transition path, we have no unique answer regarding tridymite (the least studied stable silica phase under pressure due its prolific polymorphism even at room temperature and pressure, we are working on it). However, given that cristobalite and trydimite are the most common silica polymorphs in most planetary materials, particularly in meteorites, we believe that quartz was unlikely the precursor in any of the meteorites where seifertite was observed.

Q7 “Line 201-202: Again, heterogeneous temperature distribution in shocked meteorites should be considered.”

A7: If we understood Reviewer #3 correctly, the idea is that due to temperature gradients cristobalite may crystallize from silica melt and co-exist with stishovite and seifertite. If so, such a scenario has already

been considered and refuted in literature (see, for example, for Martian meteorites Shaw and Walton, *Meteoritics and Planetary Sciences* 48,758 (2013): “Considering all of the model calculations, even smallest isolated pockets have cooling times greater than duration of the pressure pulse. The crystallization products of these shock melts will be unrelated to the peak shock experienced by the meteorite”).

Q8 “Line 212-213: Based on Kubo et al. (2015), it depends on the duration of P and T.”

A8: We are not de-emphasizing possible role of kinetics (Ref. 11), but just underline a possible different mechanism and provide the crucial missing piece of information to better understand the kinetics data presented previously – that cristobalite X-I is comprised of SiO₆. Note that according to Kubo et al. co-existence of seifertite and stishovite is possible, but will require very specific combination of conditions (including impactor size) thus suggesting that finding simultaneously seifertite and stishovite would be not a common phenomenon (which in fact is opposite – in all known cases seifertite was found together with stishovite). Moreover, Kubo et al. do not explain co-existence of high-pressure phases with cristobalite, and we now could re-interpret consequences of their experiments: they observed region of co-existence of cristobalite X-I, stishovite, and seifertite, and decompression of this assembly will result in mixture of α -cristobalite, seifertite, and stishovite as found in natural samples. Further detailed experiments and investigations of natural samples may further clarify the situation.

Q9 “Line 228-230: Terrestrial silica phase is mainly quartz. Quartz behaves like α -cristobalite? Your explanation is needed here.”

A9: No, it obviously does not behave like alpha-cristobalite. We provide enough references on high-pressure behavior of quartz.

Q10 “Others: The authors should distinguish your opinions (or interpretations) from results if you prefer to use both "results" and "discussion" sections. For example, you mention that "We suspect that electron irradiation might cause short-range transformation of unstable seifertite to the more stable α -cristobalite." in results section (line 189-191). This should be included in your discussion section.”

A10: We are fully agreed with Reviewer #3 and follow this rule as much as we can. However, in some situation we believe it is in the readers’ interest to get some comment immediately rather than search for explanations few pages later.

Reviewers' Comments:

Reviewer #1:

Remarks to the Author:

The authors have clarified several of the issues emerged in the former round of reviews.

While I am satisfied with the overall quality of the manuscript at this stage, I still find the sentence added about LDA unsatisfactory. The authors should clarify that LDA, not only gives wrong results for quartz, but it gives completely wrong transition pressures for all the tetrahedral phases turning into octahedral phases.

While I find acceptable and well justified the use of LDA for structural and vibrational properties of octahedral phases, it would be very useful and not particularly computationally expensive to plot the E(V) curves of alpha-Cristobalite, Cristobalite XI and Seifertite, computed with a more accurate generalized gradient approximation functional (e.g. PBE). Such calculation would provide a theoretical estimate of the transition pressures, which is missing from the current picture.

Hence I recommend that this work is published in Nature Communications, provided that the authors take into account my suggestion for improvement of the theoretical modeling.

Reviewer #2:

Remarks to the Author:

Report on the revised manuscript (101404_1_art_file_2104506_ppm1fr) entitled "Compressional pathway of α -cristobalite and structure of cristobalite X-I" by A. Cernok, K. Marquardt, R. Caracas, et al.

The authors have revised the submitted manuscript in a very thoroughly way, large parts have been rewritten and the title changed. Honestly, I am a bit surprised how consequently the authors followed the suggestions of the reviewers, I would have even advocated a title of something like "Compressional pathway of α -cristobalite, structure of cristobalite X-I and its potential role towards natural seifertite formation".

To my opinion, the resulting work present now straightforward and highly concise a huge amount of challenging data, which are focused on the main results, largely the characterisation of the structure of cristobalite X-I, its importance in the transformation path-way of α -cristobalite and the importance of non-hydrostatic conditions on the resulting phases assemblage. In short, I recommend hence the presented work as being ready for publication.

Following the rebuttals point by point:

Q1: The manuscript has been largely modified and point out the most interesting and new results clearly – good work.

Q2: In the revised work, the discussion is by far more straightforward, avoiding questionable statements reminding to be only „half-truths“. I agree, the explanations concerning the word "paragenesis" are correct, but it is probably good to avoid that term, because I remember, that the term has been used particularly in a thermodynamically way to pronounce the contrariety in peak shock pressures in the past. All further arguments are correct (I confirm the opinion of A. El Goresy from own discussions with him) and I see no further as critical or ambiguous points in the manuscript.

Q3: The manuscript benefits from the added literature on former structural models of cristobalite X-I. I have no doubt, the structural solution seems reliable, due to the profound way of data acquisition and particularly because single crystal data are used this time. I was just wondering how so different models could appear so similar in powder diffraction. But since differences occur

even in d-spacings, no further comments are necessary.

Q4: Since random orientations did not change transformation pressure, the presented data are sufficient. It was not my purpose to stimulate a detailed study on transformational stress.

Reviewer #3:

Remarks to the Author:

The authors adequately replied to the issues I raised.

Their finding will bring a clue for understanding the curious behavior of silica, which is significant not only for high-pressure mineralogy but also for shocked meteorites.

Reviewer 1:

- GGA calculations have been performed and we added E-V curves as an inset in Figure 4 and introduced a new Supplementary Figure 8 on GGA data
- We have added some sentences to expand text on GGA calculations in the Results and Method sections (highlighted in yellow)
- Having expanded our manuscript significantly with theoretical data, Dr. Caracas was introduced as the second corresponding author, because he performed the calculations himself

Reviewer 2:

- Following advice of Reviewer 2 we have extended our Title to address the seifertite formation